# Conformational dynamics in TRPV1 channels reported by an encoded coumarin amino acid

Ximena Steinberg[1†], Marina A Kasimova[2†], Deny Cabezas-Bratesco[1], Jason D Galpin[3], Ernesto Ladron-de-Guevara[4], Federica Villa[5], Vincenzo Carnevale[2], Leon Islas[4], Christopher A Ahern[3], Sebastian E Brauchi[1,6]*

[1]Physiology Department, Faculty of Medicine, Universidad Austral de Chile, Valdivia, Chile; [2]Institute for Computational Molecular Science, Temple University, Philadelphia, United States; [3]Department of Molecular Physiology and Biophysics, University of Iowa, Iowa City, United States; [4]Departamento de Fisiología, Facultad de Medicina, Universidad Nacional Autónoma de México, Ciudad de México, Mexico; [5]Dipartimento di Elettronica, Informazione e Bioingegneria (DEIB), Politecnico di Milano (POLIMI), Milano, Italy; [6]Millennium Nucleus of Ion Channels-associated Diseases (MiNICAD), Iniciativa Cientifica Milenio, Santiago, Chile

*For correspondence:
sbrauchi@uach.cl

[†]These authors contributed equally to this work

Competing interests: The authors declare that no competing interests exist.

**Abstract** TRPV1 channels support the detection of noxious and nociceptive input. Currently available functional and structural data suggest that TRPV1 channels have two gates within their permeation pathway: one formed by a 'bundle-crossing' at the intracellular entrance and a second constriction at the selectivity filter. To describe conformational changes associated with channel gating, the fluorescent non-canonical amino acid coumarin-tyrosine was genetically encoded at Y671, a residue proximal to the selectivity filter. Total internal reflection fluorescence microscopy was performed to image the conformational dynamics of the channels in live cells. Photon counts and optical fluctuations from coumarin encoded within TRPV1 tetramers correlates with channel activation by capsaicin, providing an optical marker of conformational dynamics at the selectivity filter. In agreement with the fluorescence data, molecular dynamics simulations display alternating solvent exposure of Y671 in the closed and open states. Overall, the data point to a dynamic selectivity filter that may serve as a gate for permeation.

## Introduction

Cation channels from the TRPV family are functional tetramers with a central pore domain that is shaped by TM5 and TM6 transmembrane segments from each of the four subunits (*Ramsey et al., 2006*) (*Figure 1a*). As in *Shaker*-related tetrameric ion channels, the pore domain houses the ion selectivity filter and an intracellular 'bundle crossing' formed by the α-helical S6 segments. Functional and structural data suggest that the selectivity filter and inner bundle crossing have the potential to control the flux of ions according to the electrochemical gradient (*Cao et al., 2013*; *Gao et al., 2016*; *Salazar et al., 2009*). The lower constriction is highly conserved (*Palovcak et al., 2015*) and associated to the canonical gate (*Cao et al., 2013*; *Steinberg et al., 2014*), but less is known about the putative role of the upper constriction at the ionic selectivity filter, possibly due to the functionally sensitive nature of this region to side-chain mutagenesis. Nevertheless, high-resolution structures of TRPV1 suggest that the upper constriction undergoes structural changes in the presence of DkTx (*Gao et al., 2016*), a spider toxin that increases the open probability of the channel (*Siemens et al., 2006*). Moreover, the inhibition of the unitary conductance of TRPV1 channels by protons suggests that the selectivity filter region is flexible and is capable of adopting multiple,

**eLife digest** Cells use proteins on their surface as sensors to help them to assess and explore their environments and adapt to external conditions. The transient receptor potential (TRP) ion channels form one such family of proteins. Sodium, potassium and calcium ions can move through TRP channels to enter and exit cells, and by doing so trigger changes in the cell that help it respond to an external stimulus. The channels have physical "gates" that can open and close to control the flow of the ions. When the TRP channel detects a stimulus – which could take the form of specific chemicals, or a change in temperature, pressure or voltage – it changes shape, causing the gate to open.

Researchers have a number of unanswered questions about how TRP channels work. Where in the channels are gates located? How do the channels control the flow of ions, and how do they communicate with each other? And which regions of the protein sense environmental cues? As a result, new technologies are being developed to make it easier to study the types of rearrangements that TRP channels experience when they activate.

Steinberg, Kasimova et al. have used total internal reflection microscopy – a method that images fluorescent molecules – to measure the conformational change of a single TRP channel in a living cell. This channel, called TRPV1, senses external heat and plays an important role in pain perception. Its gate can also be opened by the pungent compound of chili pepper, capsaicin.

The results of the experiments suggest that a selectivity filter region in TRPV1 channels changes its shape when the channel opens in response to capsaicin. Then, this selectivity filter appears to do double duty – it controls which types of ions pass through the channels as well as controlling their flow rate.

Because of its role in pain perception, having a better understanding of how TRPV1 works will be valuable for researchers who are trying to develop new pain relief treatments. The so-called 'seeing is believing' method used by Steinberg, Kasimova et al. could also be used to study other membrane proteins, both to guide drug development and to improve our understanding of how cells interact with their environment.

functionally distinct conformational states (*Liu et al., 2009*; *Wen et al., 2016*). However, it is not known if such conformational dynamics at the selectivity filter might be coupled to channel activation by agonists, such as capsaicin, and if filter gating is sufficient or acts in concert with the lower bundle crossing to control the ionic conductance.

Organic dyes with environmentally sensitive fluorescence have been used for nearly 20 years to describe membrane protein conformational changes and ion channel motions (*Mannuzzu et al., 1996*). The voltage-clamp fluorometry (VCF) approach relies on the covalent attachment of such fluorophores, usually via introduced cysteine residues, and allows for the simultaneous recording of fluorescence signals and voltage-clamped ionic currents from expressed ion channels, often in the *Xenopus* oocyte (*Cha and Bezanilla, 1997*). This technique has proven to be useful for the real-time description of subtle, and in some cases electrophysiologically silent, conformational changes in voltage- and ligand-gated ion channels (*Cha et al., 1999*; *Islas and Zagotta, 2006*; *Pless and Lynch, 2009*). However, excessive background labeling of intracellular sites, at endogenous cysteine residues for instance, and the prerequisite for solvent accessibility of the labeling site serve to limit the utility of the technique substantially for the study of intracellular or solvent-restricted residues. A possible solution to these limitations is the use of genetically encoded fluorophores in the form of non-canonical amino acids (f-ncAA) (*Drabkin et al., 1996*). Such strategies employ an orthogonal suppressor tRNA and an evolved tRNA synthetase (RS), which can be used to encode the ncAA at any site in the reading frame of the target gene (*Drabkin et al., 1996*; *Sakamoto et al., 2002*). This approach has been used for site-specific incorporation of fluorescent ncAAs on soluble proteins in prokaryotic systems (*Summerer et al., 2006*; *Wang et al., 2006*), membrane proteins expressed in *Xenopus* oocytes (*Kalstrup and Blunck, 2013*;), and proteins expressed in mammalian cells (*Chatterjee et al., 2013*; *Luo et al., 2014*; *Shen et al., 2011*; *Steinberg et al., 2016*; *Zagotta et al., 2016*). The f-ncAA coumarin has potential for the study of expressed protein conformational dynamics because of its small size and its exquisite sensitivity to environmental polarity

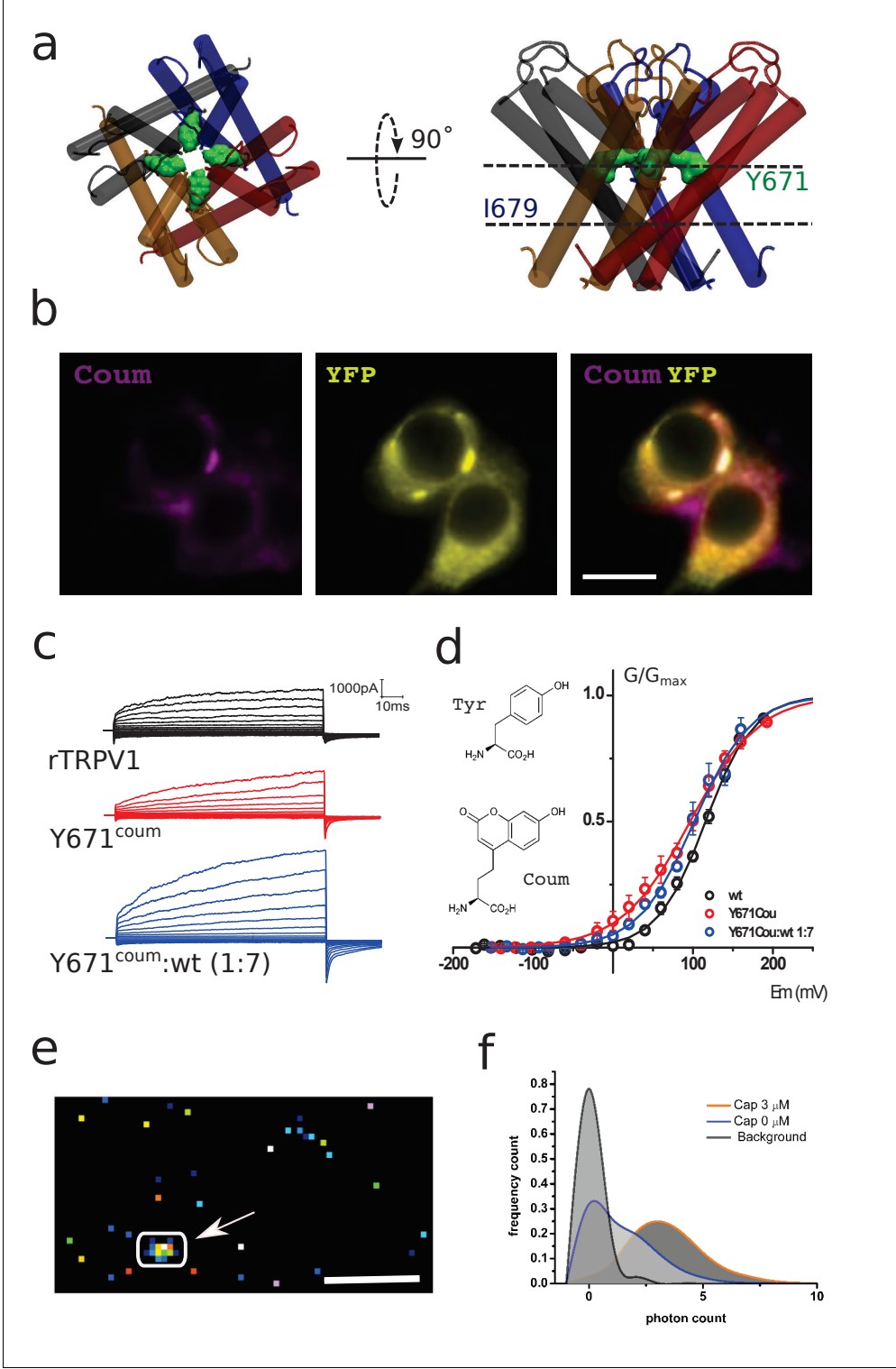

**Figure 1.** Functional expression of coumarin-containing TRPV1 channels at the membrane of HEK-293T cells. (a) Top-down and lateral view of the pore region of rTRPV1 tetramers. Dashed lines denote the positions of putative gates, a canonical gate at I679 and a secondary constriction within the pore at Y671 (green shade). The different subunits are shown in different colors. (b) Epifluorescence images obtained from transiently transfected HEK-293T cells. Cells were co-transfected with wild-type rTRPV1-YPF and with rTRPV1-Y671$^{coum}$. The scale bar corresponds to 10 µm. Images correspond to the average of 50 frames. (c) Representative traces obtained from transiently transfected HEK-293T cells subjected to voltage steps from −180 mV to +180 mV, in 20 mV increments,

*Figure 1 continued on next page*

*Figure 1 continued*

for the three different conditions indicated. No TRPV1 currents were seen from expressed Y671TAG channels co-expressed with the coumarin Rs and tRNA but lacking the supplemented amino acid in the culture media. (d) Conductance-voltage relation for the wild-type rTRPV1 (black) and the two different coumarin-containing channels (TRPV1-coumarin alone in red; 1xTRPV1coumarin +7xTRPV1wt in blue). Curves were fitted to a Boltzmann distribution of the form $I = I_{max}/(1 + \exp([zF(V-V_{0.5})/RT])$, where z is the voltage dependency, $V_{0.5}$ is the half-activation voltage, and $I_{max}$ is the maximum current. Each curve represents the average of at least six different experiments performed at 20°C. Error bars correspond to standard errors of the mean (SEM). The structure of coumarin and tyrosine are depicted in wireframe. (e) Ten thousand frames were averaged and a threshold was set on the averaged image in order to define regions of interest (arrow indicates ROI). The scale bar corresponds to 5 μm. (f) The photon count was extracted from the original set of frames. Capsaicin incubation promotes a higher photon count distribution.

The online version of this article includes the following figure supplement(s) for figure 1:

**Figure supplement 1.** Two composites of epifluorescence images obtained from transiently transfected HEK-293T cells.

**Figure supplement 2.** Capsaicin induces an increase in the photon count.

---

(*Liu et al., 2015*; *Wagner, 2009*; *Wang et al., 2006*). This sensitivity can be exploited to report on variations (i.e. dielectric changes) in the surroundings of the incorporated f-ncAA and, in turn, it can be interpreted as a signature of a protein conformational change. Notable examples include the use of coumarin and the amber codon suppression technique to report on peptide binding to viral proteins (*Ugwumba et al., 2011*), on the photoregulation of firefly luciferase and on the subcellular localization of proteins in living mammalian cells (*Luo et al., 2014*), as well as onstructural rearrangements in isolated NaK channels (*Liu et al., 2015*).

Here, the spectral dynamics of an encoded hydroxy-coumarin were determined in individual live cells with total internal reflection (TIRF) microscopy and rapid imaging of single emitters to study the conformational changes occurring within the pore of the capsaicin receptor TRPV1 during activation.

## Results

### Expression of Tyr-coumarin at genetically encoded positions

Previous cysteine accessibility experiments showed that Y671C residues define the boundary for both silver ($Ag^+$) state-dependent accessibility and disulphide bond formation in the closed state (*Salazar et al., 2009*). Thus, we reasoned that a genetically encoded coumarin residue at position Y671 ($Y671^{Coum}$) would be functionally tolerated and if so, could potentially be well positioned to report on local structural rearrangements occurring within the pore during gating (*Figure 1a*). As expected for cells containing an encoded fluorescent probe, HEK-293T cells heterologously expressing both TRPV1-YFP and amber codon-containing TRPV1 mutants ($TRPV1^{YFP}/TRPV1^{TAG}$) exhibit coumarin–YFP colocalization (*Figure 1b*; *Figure 1—figure supplement 1*) consistent with the presence of coumarin and YFP co-fluorescence. Whole-cell patch-clamp electrophysiological recordings performed at room temperature (22°C) demonstrate the functional expression of the coumarin-containing TRPV1 channels ($TRPV1^{TAG}$) when expressed alone or at 1:7 molar ratio ($TRPV1^{TAG}$:TRPV1–YFP) (*Figure 1c*). The later condition was chosen to simplify the imaging of single emitters (i.e. one coumarin molecule per channel tetramer or cluster of tetramers). Both macroscopic channel kinetics and the conductance-voltage (G-V) relationship were similar between wild type (WT) and coumarin-containing channels (*Figure 1c*). The G-V relationship from wild type and coumarin channels were well fit by a Boltzmann function with a half-maximal activation voltage ($V_{0.5}$) near +100 mV (*Figure 1d*). Thus, the data suggest that position Y671 in TRPV1 is amenable to nonsense suppression and encoding of f-ncAA coumarin.

### Optical recordings of capsaicin-induced changes in solvation

In order to demonstrate the ability of the genetically encoded hydroxy-coumarin side chain to act as a reporter for protein activity in live cells, HEK Y671-coumarin TRPV1 expressing cells were imaged within the evanescent field in total internal reflection fluorescence (TIRF) microscopy, with a single-photon counting avalanche diode (SPAD) camera (64 × 32 pixels array with in-pixel counters)

(*Bronzi et al., 2014*; *Michalet et al., 2013*). Specifically, photon counts of the emission signal were examined to determine the effect of capsaicin, the canonical agonist of the TRPV1 channel (*Figure 1e and f*; *Figure 1—figure supplement 2*). The photon count distribution showed clear differences between pixels representing background and pixels from fluorescent spots, which presumably arise from fluorescent channel subunits (*Figure 1f*). Further, when the cell is exposed to a maximal concentration of capsaicin (3 μM), the photon counting rate further shifted to higher values (*Figure 1f*). The modes observed for these populations showed significant differences (n = 4; p<0.05). One interpretation of the capsaicin-induced increase in photon emission counts is that it stems from local changes in the immediate environment of the dye. To simplify the analysis and interpretation of the optical data, we set out to measure the fluctuations of the fluorescence signal emitted from a single diffraction-limited emitter directly. Given the large pixel size and the low fill-factor of the SPAD array used (*Bronzi et al., 2014*; *Michalet et al., 2013*), the precise localization of a diffraction-limited signal is not possible under the condition of overexpression of membrane proteins in living cells. Therefore, steady state fluctuations of fluorescence were recoded by an electron-multiplying charge coupled device (EM-CCD) camera to resolve changes in the optical properties of the dye (*Blunck et al., 2008*).

As in the electrophysiological experiments, cDNAs encoding for both wild-type channels (TRPV1) and amber-codon-containing mutants were mixed in 7:1 ratio to promote the formation of uneven channel tetramers. This strategy results in a small yet discrete number of diffraction-limited coumarin-emitting spots as observed by TIRF microscopy (*Figure 2a*). The coumarin signal colocalizes well with the YFP signal corresponding to wild-type YFP-tagged channel subunits (*Figure 2a*).

Close examination of these fluorescent spots by photobleaching showed a stepwise behavior allowing for ex-post identification of spots containing a single emitter (*Figure 2b*). A clear change in fluorescence was evidenced by the ensemble average of several diffraction-limited spots once the cell is exposed to capsaicin (3 μM; *Figure 2c*). The ensemble recording showed not only an increase

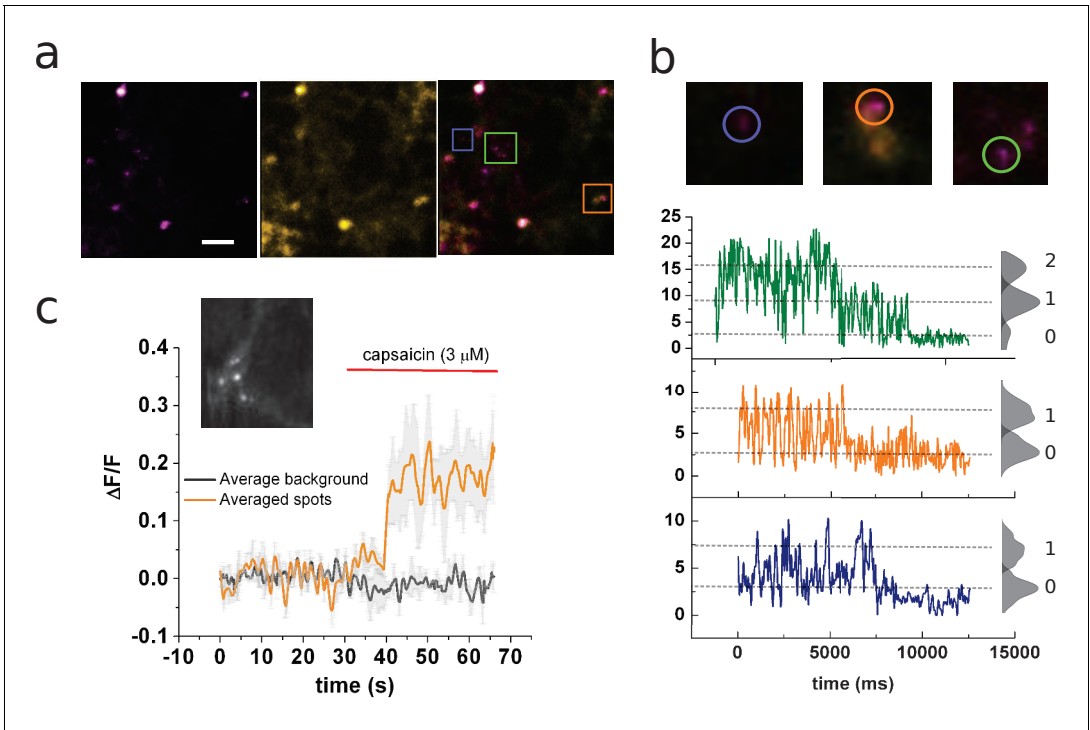

**Figure 2.** Identification of channels expressing a single coumarin emitter. (a) Representative TIRF images of the 7:1 transfected cells. Images correspond to the average of 100 frames. Colocalization of coumarin diffraction-limited puncta with YFP signal is marked by colored squares. Scale bar corresponds to 2 μm. (b) Representative traces of photobleaching coumarin-positive spots. Top images correspond to the ROIs depicted in (a). Color-coded traces depict the time course of the fluorescence signal used later for the identification of individual spots that bleach on a single step. (c) Ensemble average of the signal recorded from multiple spots from the cell's membrane shown in the inset. The image corresponds to the average of 80 frames taken on the coumarin channel (ex405nm/em450nm).

of fluorescence after agonist treatment but also increases in the standard deviation of the signal's mean value (*Figure 2c*).

## Activation kinetics obtained from the optical signal

Given the Y671 position within the TRPV1 selectivity filter, we reasoned that the environmental sensitivity of coumarin may well provide kinetic information on channel gating. Stationary noise analysis was performed on the basis of the autocorrelation ($C_{(\Delta t)}$) of the fluorescence signal (*Figure 3a*). The autocovariance function can be used to calculate the correlation between values of a noisy signal taken at increasing time intervals, $\Delta t$. In this regard, $C_{(\Delta t)}$ will be higher for small values of $\Delta t$ (i.e. for data points close to each other) and this function rapidly decays as $\Delta t$ become larger. In the present example, this analysis allows for the identification of a characteristic time constant ($\tau$) of the exponential decay, which describes the autocorrelation of the fluorescent signal originating from single emitters. The obtained $\tau$ has an inverse relationship to the rates defining the equilibrium of the reaction between open and closed states (*Anderson and Stevens, 1973*; *Zingsheim and Neher, 1974*). The calculated autocorrelation function discriminates well between the three conditions tested (i.e background, no capsaicin, and capsaicin present in the media). The background signal is obtained from cells that have the coumarin amino acid but lack the expressed TRPV1$^{TAG}$ cDNA. This condition generated noise with a single, fast time constant ($\tau$ = 8.5 ms) that was easily separated from the double exponential decay time constants observed in the signal originating from the encoded coumarin conditions. In the absence of capsaicin, the coumarin signal autocorrelation had two time constants with corresponding values of 12 ± 6 ms and 78 ± 22 ms. Incubations with saturating capsaicin concentrations (3 µM) changed these time constants to 13 ± 4 ms and 148 ± 38 ms (n = 7; *Figure 3a*), consistent with the possibility that the encoded coumarin is reporting on an agonist-induced increase in $\tau$. Such a result can be interpreted as an stabilization of the bright state after ligand binding (i.e. longer bursts of activity) either alone or in combination with the destabilization of the dark or non-emitting state, a possibility consistent with the previously determined distribution of closed and open states in TRPV1 channels when activated by capsaicin (*Hui et al., 2003*).

In order to begin to distinguish between these possibilities, we examined unitary transitions in the optical data (*Figure 3b*; see Materials and methods). To do so, we inferred that the signal minima (i.e. $L_0$) would correspond to the magnitude of background signal and would corroborate that fluorescence transitions drop from a given maxima ($L_1$) to the observed minima ($L_0$) as expected (*Figure 3b*, inset). We analyzed the signal in the presence of capsaicin using a threshold-crossing criterion, where the threshold was chosen as 2.5 times the standard deviation of the mean amplitude of background fluctuations (see Materials and methods). This strategy produced an idealized recording, similar to the analysis of single ion channel current idealization trace (*Figure 3b*). An analysis of this record shows that the agonist does not affect the amplitude of the fluorescence fluctuations, but rather it promotes a shortening of the mean dwell-time of the bright state, which we defined as $L_1$, going from 398 ± 133 ms to 156 ± 52 ms (n = 5; *Figure 3c and d*). On the other hand, capsaicin increases the probability of being in the bright state (*Figure 3*). Taken together, these two observations support the notion that capsaicin increases the burst length of transitions to the bright state. This is in agreement with the autocorrelation data, which show that capsaicin increases the time constant, and consistent with an increased burst length of bright events. The data from single emitters also supports previous data showing that the main gating effect of capsaicin is increasing the burst-length of openings by shortening the closed state, instead of prolonging the duration of the open state (*Hui et al., 2003*).

Overall, the fluorescence data is consistent with the possibility that coumarin at position Y671 reports on the transition between open and closed states of the channel. Further, given the physicochemical properties of coumarin (i.e. that it is quenched in an aqueous environment), the data suggest that position Y671 is more water-exposed and thus has decreased in fluorescent output, in the closed state compared to the open state (*Wagner, 2009*). By comparison, the available TRPV1 structural data place W426 in an apparently invariant solvent-exposed region (*Figure 3—figure supplement 1a*), thus, the open-closed transition would not be expected to induce evident changes in local solvation (*Gao et al., 2016*). In order to investigate the specificity of the spectral effects measured at Y671 with coumarin, a second site, W426, was examined in parallel. Electrophysiological analysis confirmed that W426$^{coum}$ is a functional channel (*Figure 3—figure supplement 1b and c*), and the photon-count analysis suggests that the fluorescence signal from this position is insensitive to

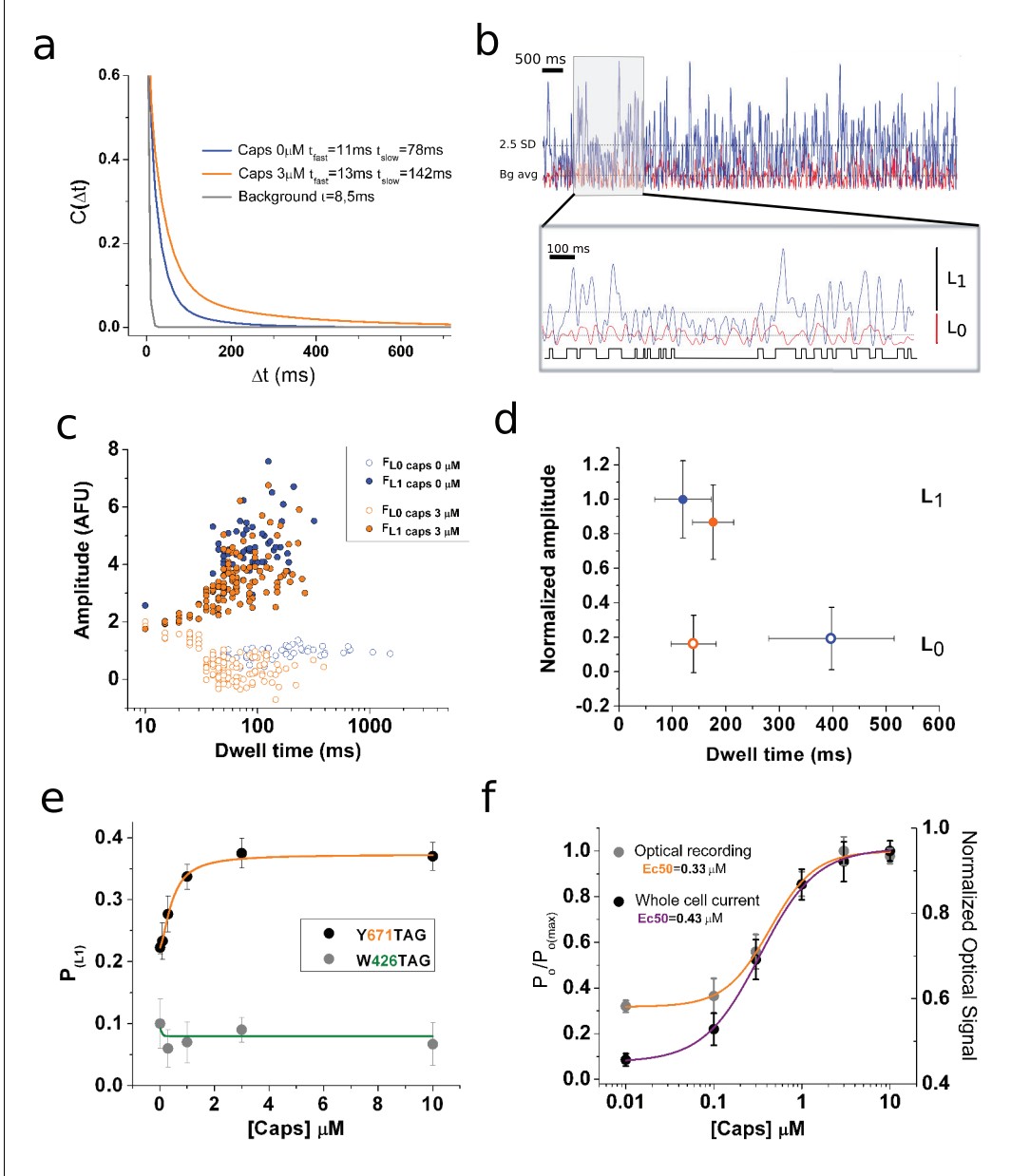

**Figure 3.** Optical studies of ion channel gating. (**a**) Stationary noise of single spots, with each curve corresponding to the averaged fitted curves for the indicated conditions (n = 7, paired). The autocorrelation function was obtained for background (gray), capsaicin free (blue), and 3 μM capsaicin (orange). (**b**) Analysis of fluorescence fluctuations. Superimposed traces of capsaicin (blue) and background (red). The threshold for level one is highlighted (2.5 standard deviations [SD], dotted line). The gray box indicates the zoomed-in portion of the recording presented in the inset below, with the dotted lines indicating background average.. An idealization is shown in black. (**c**) Semi-log plot of the amplitude versus dwell-time shows a shortening of mean time $L_0$. (**d**) Plot summarizing datasets of amplitude versus dwell time at 0 and 3 μM capsaicin (n = 5, paired). Error bars correspond to SEM. (**e**) Dose–response curve showing changes in the probability of level 1 ($P_{L1}$), calculated from non-corrected fluorescence fluctuations, for the two positions tested for the indicated channel types expressing coumarin. (**f**) Normalized dose–response curves for background subtracted fluorescence fluctuations (orange) and whole-cell electrophysiological response (purple). The electrophysiological response was tested at −70 mV in the presence of the indicated concentrations of capsaicin. Error bars correspond to SEM.

The online version of this article includes the following figure supplement(s) for figure 3:

**Figure supplement 1.**

**Figure supplement 2.** Distribution of the whole data set obtained from the fluctuation analysis of the fluorescence signal.

capsaicin administration (*Figure 3—figure supplement 1d and e*). Further, the fluctuations of coumarin fluorescence at $426^{coum}$ are unresponsive to capsaicin treatment, as determined by the calculated bright state probability, $P_{L1}$ (*Figure 3e*; *Figure 3—figure supplement 1f and g*). Conversely, the $P_{L1}$ of position $671^{coum}$ reported capsaicin dose-dependent fluorescent changes which are well fitted by the capsaicin dose-response curve obtained from capsaicin-activated currents at −70 mV (*Figure 3e*; *Figure 3—figure supplement 2*). Lastly, when the fluorescence data from $Y671^{coum}$ was normalized and compared with the electrophysiological data taken from whole-cell macroscopic recordings, we observed a close match between the calculated $Ec_{50}$ for optical and electrophysiological data, 0.33 µM versus 0.43 µM, respectively (*Figure 3f*). Taken together, the data and biophysical analysis suggest a continuum of coordinated motions within the pore, with the $Y671^{coum}$ optical data reporting on a transition to the open conducting conformation in the selectivity filter.

## Rearrangements of tyrosine 671 during closed-open transition

To relate the changes in coumarin fluorescence to possible structural transitions experienced by the channel during gating, we analyzed the molecular dynamics (MD) trajectories of wild-type and $671^{coum}$ TRPV1 channels in the closed and open states (about 750 ns each). First, we compared the conformations that Y671 adopts in the closed and open states of TRPV1 (*Figure 4a and b*).

In the closed state, the representative conformation shows that three out of four phenyl moieties are approximately perpendicular to the membrane and do not contact each other (*Figure 4a*). In this conformation, phenyl moieties create transient hydrogen bonds with the main-chain of F640 of the selectivity filter. In the open state, the planar groups of the four tyrosine residues instead adopt a parallel orientation to the membrane, forming a ring around the central pore (*Figure 4b*). These opposing orientations in the closed and open conformations predicts an increase in the hydration of the central pore, with the four Y671 moieties being more hydrated in the closed state (*Figure 4c*). Surprisingly, the open state shows an intermittent occupancy by water molecules of the region surrounded by the four Y671 side chains. Despite being somewhat counterintuitive, this observation is not in contradiction with the conductive nature of the open state: even when devoid of water molecules, the environment provided by the ring of hydroxyl groups from Y671 is highly polar and does not impede ion permeation. In spite of these water density fluctuations, the pore is, on average, continuously hydrated only in the open state (*Figure 5a*). $Na^{+}$-permeation events in our simulations (in the absence of any applied external field) were observed in the open state, but not in the closed state.

An additional insight provided by the simulations is that the hydration of Y671 is different in the two conformations: in the open channel, one face of the phenyl ring is exposed and the adjacent ring is partially buried by F640 and M644, whereas in the closed state, both faces of the Y671 side chain are exposed to water. To describe hydration of Y671 quantitatively, we calculated the corresponding solvent accessible surface area (SASA) in the open and closed conformations. The evolution of the Y671 SASA reveals that the two MD trajectories start to diverge quite early: in particular, in both conformations, the Y671 SASA drops down from ~35% to ~20% at the beginning of the equilibration; then, after ~30 ns, in the closed and open states, respectively, the SASA either increases up to ~30% or continues to decrease until it reaches an equilibrium value of ~15% (*Figure 4d*). Further, it can be observed that although the open state displays the side chains of Y671 residues engaged in mutual hydrogen bonding interactions, they establish vdW interactions with nearby residues F640 and M644, whose side chains are in between the selectivity filter backbone groups of two adjacent subunits (*Figure 4e*). In the closed state, F640 and M644 residues still interact with Y671 as in the open state; however, the different orientation of the side chain phenyl groups allows for a rigid displacement of the pore helix toward the extracellular side of the pore, narrowing the diameter of the selectivity filter (*Figure 4a and e* and *Figure 5a and b*).

To explore the plausibility of this hypothetical gating mechanism, we performed additional simulations in whichfour coumarin moieties were modeled in the place of the side chains of Y671. The rationale for these additional calculations was to ascertain whether the bulkier coumarin moiety (as compared to tyrosine phenyl group) can accommodate configurations similar to those accomodated by the wild type. In particular, would the 'ring configuration' in the open state allow ion permeation in the case of coumarin? Is coumarin in the 'vertical configuration' able to establish H-bonds with and thus promote the displacement of the pore helix in the closed state? Although the length of the simulations performed on the coumarin mutant would need to be extended in order to

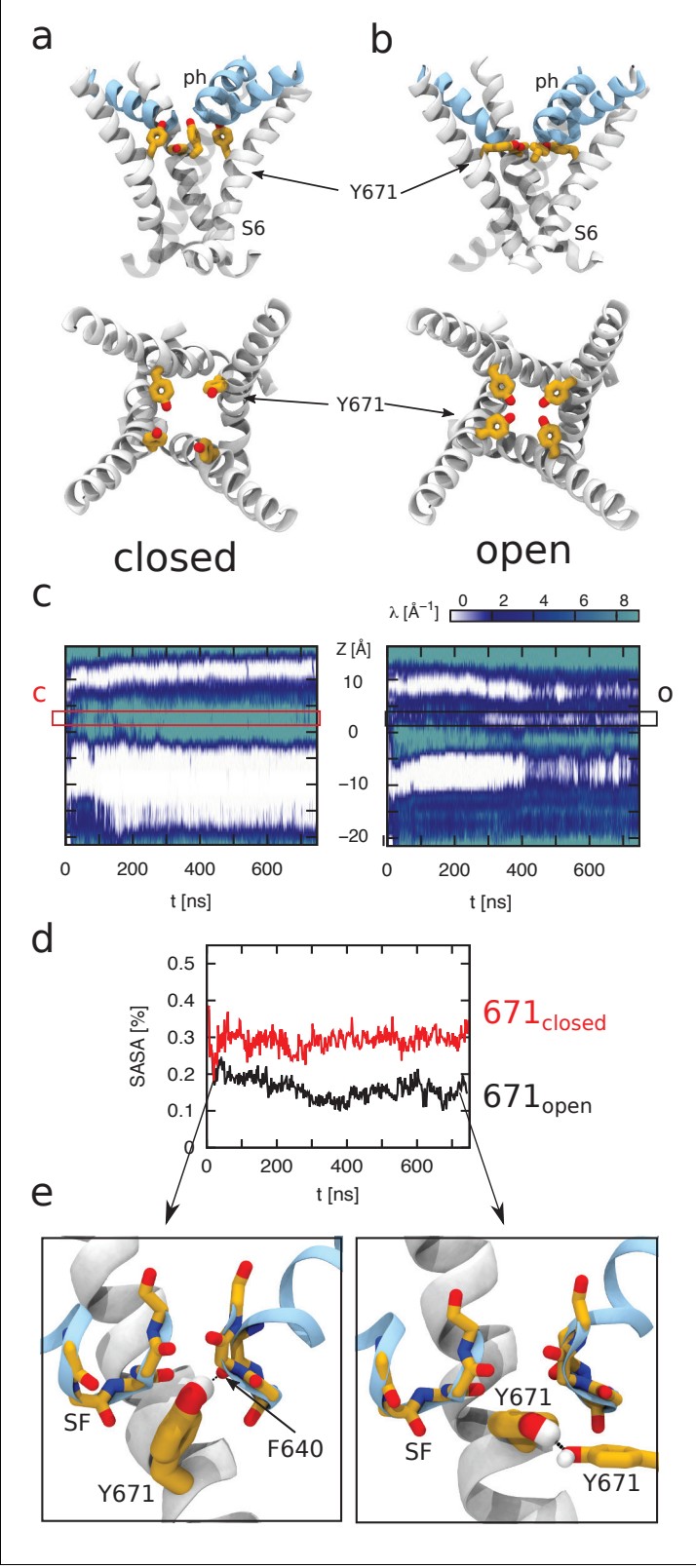

**Figure 4.** Molecular dynamics simulations to calculate hydration of Y671. Molecular dynamics (MD) simulations of TRPV1 channels in the open (a) and closed (b) states. The upper and lower panels correspond to the side and top views of the S6 helices bundle (white), respectively. The pore helices (ph) are shown in blue. Y671 residues are represented as yellow sticks. In the open state, four Y671 residues create a ring oriented parallel to the membrane

*Figure 4 continued on next page*

*Figure 4 continued*

plane (perpendicular to the pore principal axis); whereas in the closed state, three of these residues do not interact with each other and are oriented perpendicular to the membrane plane (parallel to the pore principal axis). (c) Hydration of the channel central pore. Panel c shows the time evolution of the density profile (of water oxygens) along the pore. Specifically, at each time frame, the water oxygen atoms number density (i.e. the number of particles per unit of length) is calculated as a function of z (the axis of the pore). This quantity is shown using a color scale. Note that the units are $\text{Å}^{-1}$ (the number density for a linear system has the dimensions of the inverse of a length). The left and right panels correspond to the closed and open states, respectively. The y-axis of the plot denotes the z-axis of the system (parallel to the channel central pore). The origin of the axis is set at the center of the pore domain transmembrane part. The Y671 residues are located at ~2.5 Å. Note that in the open state, the water density is often interrupted starting from ~280 ns, whereas in the closed state, this density is continuous. (d) Solvent-accessible surface area (SASA) of Y671. Black and red curves correspond, respectively, to SASA over time for the open (black) and closed (red) states as indicated. In the closed state, the Y671 SASA is approximately twice as large as that in the open state starting from ~300 ns. (e) The panels show zoomed view of the Y671 in the closed (left) and open (right) states. In the open state, Y671 establishes a hydrogen bond with the Y671 of the adjacent subunit; whereas in the closed state, this residue interacts with the backbone of F640 located at the pore helix. The backbone of the selectivity filter (SF) is shown using a stick representation.

establish a more quantitative comparison with the experimental data, these cursory calculations suggest that the proposed gating mechanism is not incompatible with the molecular size of the coumarin moiety (*Figure 5c*; *Figure 5—figure supplement 1*).

## Discussion

Ion channel gating occurs in the range of hundreds of µs and it is one of the fastest protein-dependent processes occurring within cells (*Hille, 2001*). This rapid time-frame imposes intrinsic technical barriers in the detection of a limited number of photons coming from few emitting molecules and a consequent low signal-to-noise ratio under these conditions (*Ha, 2014*). Given the design of our imaging setups and our experimental conditions (i.e. transient overexpression), we are bound to either the low (64 × 32 pixel) spatial resolution of the ns-time-scale sampling of the SPAD imager (which does not allow us to discriminate single emitters) or to a low signal-to-noise ratio restricting us to image at a maximum of 500 Hz (ms-time scale) for the individual diffraction-limited spots observed by an EM-CCD camera. As reported previously (*Blunck et al., 2008*), we observed that the latter approach is useful for conformational transitions in the order of 2–5 ms, thus barring optical access to the fast intra-burst channel activity that occurs in tens of µs. Therefore, the unitary transition analysis presented here was limited to the description of each burst as a single opening at the expense of missing activity during short openings. None the less, such an analysis, although lacking detail on unitary transitions, is a valid approach for estimating the duration of the mean open and closed state for a ligand-gated channel such as TRPV1, which has an open transition lasting for several hundreds of ms (*Hui et al., 2003*). The ideal experimental system would allow for sparse distribution (every ~1 micrometer) of the emitter-expressing receptors at the plasma membrane, so that the photon-count profile of individual molecules can be obtained with the SPAD imager with nanosecond resolution. However, in the absence of such technical advances, standard biophysical approaches are in place to provide insights on unitary gating events from an expressed population of channels. Interestingly, the similar rate constants obtained by noise analysis of the optical signal and from the subsequent modeling of the unitary transitions, suggest a strong internal consistency of data. Another limitation to consider in the analysis is the intrinsic blinking of the dye. In the present case, this represents roughly 12% of the $P_{L1}$ transitions and therefore obfuscates efforts to obtain absolute values for the open probability. Our solution to bypass this issue was to linearly subtract this 'background open probability' post-hoc and to normalize the data to its maximal theoretical response. Still, even with this coarse approximation, the optical data is in surprising agreement with whole-cell electrophysiological recording of channel gating. Further, this approach allowed for the direct measurement of the steady-state activity of membrane receptors undergoing fast molecular rearrangements. Specifically, from our fluorescence recordings, we reasoned that Y671$^{coum}$ experiences a change of environment as part of the process of pore rearrangement associated to channel's opening. This possibility is supported by the dynamic range for the optical measurements

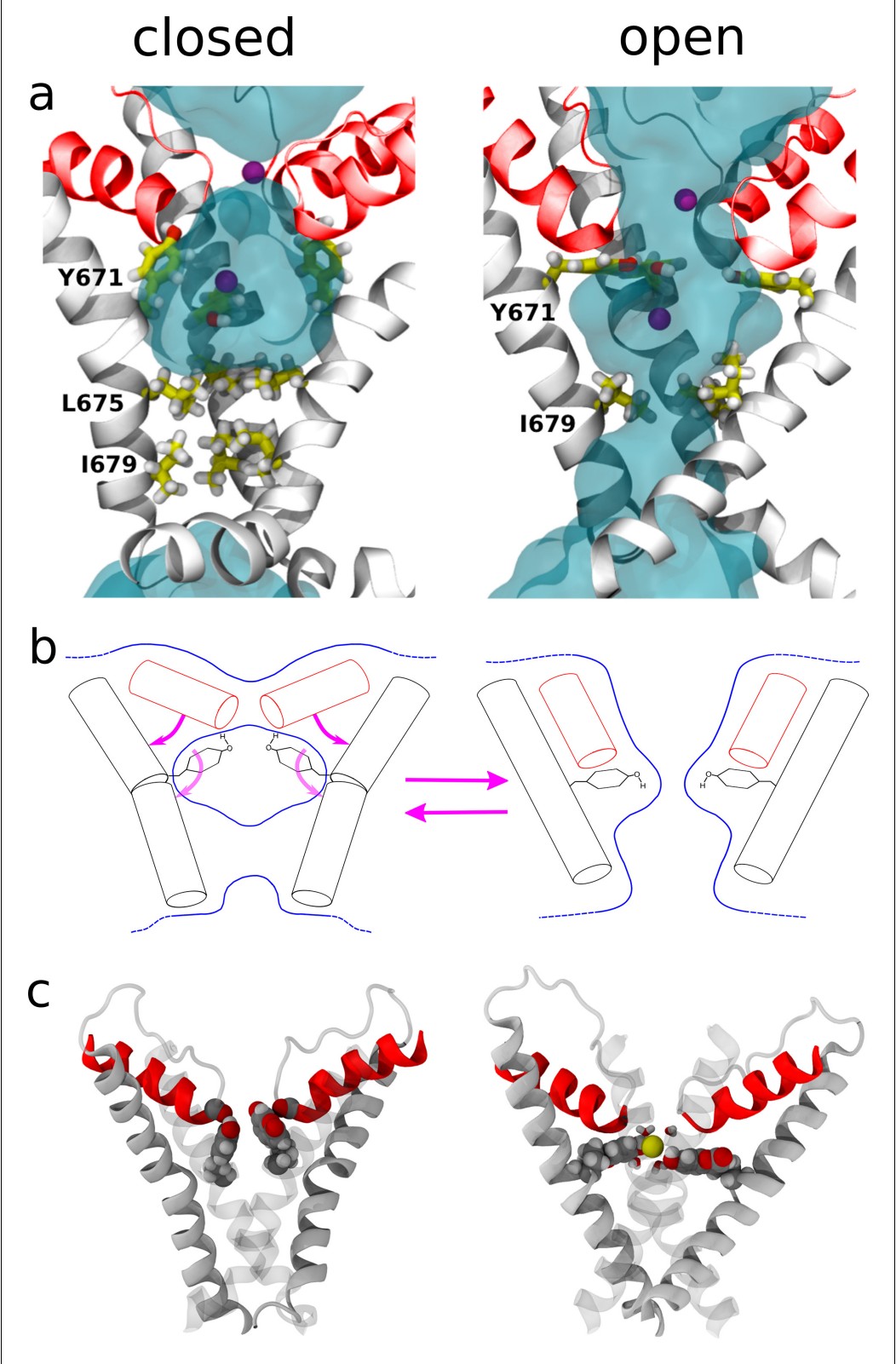

**Figure 5.** Structural changes resulting from rearrangements in the selectivity filter region. (a) Representation of the channel structures in the closed (left) and open (right) states. For clarity, only S6 (white) and pore helix (red) are shown for three subunits. The side chains of Y671 (at the upper gate) and I679, I675 (bottom gate) are highlighted using a stick representation and the permeant Na$^+$ ions are shown as purple spheres. Blue shades show the average density of water molecules calculated along the MD trajectory. Note how the water density forms a continuum throughout the pore only

*Figure 5 continued on next page*

*Figure 5 continued*

when in the open state. (**b**) Proposed mechanism for the opening and closing of the upper gate: the rearrangement of the four planar groups of residue 671 from the 'vertical' to the 'horizontal ring' configuration is accompanied by a change in the tilt (with respect to the membrane plane) of the N-terminus section of S6 and of the pore-helix. In particular, the motion of the pore-helix regulates the effective pore radius at the level of the selectivity filter. The outward motion of both the pore helix and the aromatic at position 671 cause a reduction in water and ion accessibility. (**c**) Molecular structure sampled along the MD trajectory of the coumarin mutant in the closed and open states. For clarity, only S6 and the pore helix are shown with two of the four subunits rendered as gray shading. The coumarin moieties are highlighted using a space-filling representation. Left: *closed state*. The hydroxyl group of coumarin is able to establish the same H-bonding interactions with the carbonyl group of the pore-helix observed in the wild type. Right: *open state*. A permeant $Na^+$ ion (yellow sphere) is shown together with its first solvation shell (sticks). Note how the hydroxyl groups from coumarin contribute to the shell of neighbors of the ion.

The online version of this article includes the following figure supplement(s) for figure 5:

**Figure supplement 1.** Additional views of the coumarin mutant structures.

presented here, and closely overlaps with the dynamic range of burst activity observed in single-channel recordings of TRPV1 in response to capsaicin (i.e. 0.1 to 1 µM) (*Hui et al., 2003*).

The molecular dynamics simulations showed that Y671 could undergo three significant conformational changes during the channel's transition from the conducting to the non-conducting conformation: (i) a side chain re-orientation with respect to the pore and pore helix, (ii) increased water exposure, and (iii) a reshaping of the H-bond interaction network. The spectral properties of coumarin are highly sensitive to changes in solvation and/or to changes in the coordination of the hydroxyl group of the dye (*Wagner, 2009*). Therefore, any of the modifications detected on MD simulations performed for the wild-type channel can potentially explain the observed changes in fluorescence. As a consequence, a detailed structural interpretation at the level of side-chain dynamics for the fluorescence fluctuations remains out of reach. Moreover, the data presented here do not prove a causal relationship between the motion of the aromatic side chain and the state of the pore helix.

## Materials and methods

### Cell culture

HEK293T cells (ATCC Cat# CRL-3216, RRID:CVCL_0063) were cultured in DMEM (Gibco Inc.) supplied with 10% FBS (Gibco Inc.). Plasmocin (Sigma) was used following the provider's instructions to maintain the cell culture free of mycoplasma contamination. Cells were tested for mycoplasma by standard PCR. Cells were prepared at (60–70)% confluence and transfected with lipofectamine 2000, (Life technologies). The target gene was transfected 2–3 hr after the initial transfection of the pair tRNAtag/CoumRS. The ncAA was added to the cell culture to produce a final concentration of between 0.5 and 1 µM at the time of the second transfection procedure. Under these transfection conditions, the efficiency of incorporation of non-natural amino acid was as low as 12%, with read-through calculated to be about 4% (*Steinberg et al., 2016*). Imaging: to simplify the analysis of the optical data, we transfected a mixture of TRPV1[YFP]/TRPV1[TAG] in a 7:1 ratio in an attempt to drive the system to incorporate as few coumarin molecules per tetramer as possible. The ratio of expression was not controlled as we concentrated efforts on single emitters identified ex-post. 24 hr after transfection of the target gene, cells were disaggregated and plated on poly-L-lysine treated glass covers. The f-ncAA-containing media was removed at least 12 hr prior to the experiment, allowing cells to clear the soluble ncAA. Generally, the cells were recorded 36–48 hr after the second transfection. Electrophysiology: cells were plated 2–3 hr before patching, and incubated with normal media lacking the supplemented coumarin.

### Synthesis of TyrCoum

L-(7-hydroxycoumarin-4-yl) ethylglycine was obtained as described before (*Wang et al., 2006*).

### Solutions

Ringer solution for imaging experiments contained: 140 mM NaCl, 8 mM KCl, 8 mM HEPES, and 1 mM $MgCl_2$ at pH 7.4. External electrophysiology solutions contained 145 mM NaCl, 2 mM $CaCl_2$, 5

mM KCl, 10 mM HEPES and 10 mM glucose, pH 7.4; internal electrophysiology solutions contained 135 mM CsF, 5 mM KCl, 2 mM MgCl$_2$, 1 mM CaCl$_2$, 4 mM EGTA and 20 mM HEPES, pH 7.4.

## Molecular modeling

To analyze Y671 hydration at the atomic level, we used the molecular dynamics trajectories obtained in our previous work (*Kasimova et al., 2017*). These trajectories were generated starting from the cryo-EM structure of the TRPV1 capsaicin-bound state (*Cao et al., 2013*). In one simulation, we initialized the system by inserting several water molecules (from 4 to 6) inside the channel peripheral cavities (located between the S4–S5 linker and the S6 C-terminus); in the second simulation, we left these cavities empty. During the equilibration (about 750 ns), the molecular structures converged, respectively, to the closed and open states. To estimate water density along the central pore, we implemented the following strategy. First, using HOLE software (*Smart et al., 1996*), we calculated the pore radius profile for each instantaneous configuration along the trajectory with a stride of 1 ns. Then, for the same set of frames, we calculated the three-dimensional histograms of water occupancy using the Volmap tool of VMD (RRID:SCR_001820) (*Humphrey et al., 1996*). Finally, we integrated the water occupancy in the XY plane (perpendicular to the channel central pore) using the pore radius profile as a boundary of the integration domain. The solvent accessible surface area (SASA) of Y671 in the closed and open states was estimated using the following procedure. From the MD trajectories, we extracted a set of sub-trajectories, each containing 10 frames taken with a stride of 0.2 ns. For each sub-trajectory, we computed the three-dimensional histogram of atomic occupancy (for all the atoms except water) using the Volmap tool of VMD (*Humphrey et al., 1996*). We then used this map to define a molecular surface. To this end, we first discretized the map by assigning a value of 1 or 0, depending on whether or not the local occupancy is larger than a preset threshold. We considered all the bins with a value of 1 and located 1.5 Å away from a bin with a value of 0. Finally, the Y671 SASA was calculated as the overlap between the solvent-accessible surface and the Y671 residue. Note that the SASA was normalized to the Y671 maximal surface area. Molecular dynamics simulations were performed in the open and closed states of the TRPV1 coumarin mutant. The force-field parameters for coumarin were obtained from the CHARMM general force field (*Vanommeslaeghe and MacKerell, 2012*, *Vanommeslaeghe et al., 2012*). Simulations were performed using NAMD2.10 (RRID:SCR_014894) (*Phillips et al., 2005*) at constant temperature and pressure (one atm) using the Langevin piston approach. For the vdW interactions, we used a cutoff of 11 Å with a switching function between 8 Å and 11 Å. The long-range component of electrostatic interactions was calculated using the Particle Mesh Ewald approach using a cutoff for the short-range component of 11 Å. The equations of motion were integrated using a multiple time-step algorithm, with a time step of 2 fs and long-range interactions calculated every other step. Trajectories were collected for 100 ns.

## Optical recordings

Cells were imaged using an inverted Olympus IX71 microscope main body and through-the-objective TIRF mode. Both 405 nm and 473 nm solid-state lasers (Coherent, Santa Clara, CA) were used to excite coumarin and YFP, respectively. Laser beams were focused to the backplane of a high-numerical aperture objective (Olympus 60X, N.A. 1.49, oil) by a combination of focusing lens. Fluorescence emission was collected by an Andor iXon$^{EM}$ + 860 EM-CCD camera (Andor/Oxford Instruments, Belfast, UK), after passing through an emission filter for each acquisition wavelength band of interest (coumarin: 450/70 nm; YFP:540/40 nm; Semrock, US). The light coming to the sample was controlled by a 12 mm mechanical shutter (Vincent associates, Rochester, NY), all measurements were performed under continuous light. All imaging experiments were done at room temperature (20–22°C). For *localization and co-localization*, images were recorded at 10–100 ms intervals (100–10 Hz). For *autocorrelation and unitary fluctuation analysis*, images were acquired at 2 ms intervals (500 Hz) under constant illumination at 405 nm (75% laser power equivalent to 8 mW s$^2$ mm$^2$). Laser and focus control was performed using micromanager. Acquisition and digitalization was done with Andor Solis software (Andor/Oxford Instruments, Ireland). *Photon counts* were made using a single-photon avalanche diode (SPAD) photon-counting camera (64 × 32 pixels array), which is space correlated with the EM-CCD camera, thus allowing for the location of limiting diffraction puncta. A continuous light stimulation of 1 μs duration was used to excite coumarin emitters

and photons were collected during that period. Laser TTL triggering and piezoelectric focus control (PIFOC-721, PI, Germany) was performed using Micro-Manager software (Vale Lab, UCSD, US). Acquisition and digitalization was done through a custom code written in C++ (SPADlab at POLIMI; http://www.everyphotoncounts.com/). The code used to readout the SPAD array is provided as source code in Dryad repository (https://doi.org/10.5061/dryad.1kc2c).

## Image analysis

For *localization and co-localization*, the set of frames was averaged to increase the signal-to-noise ratio (SNR=$\mu/\sigma$), which is usually less than two for the single frame at 75% laser illumination. Noise analysis of the steady-state signal by autocorrelation was performed as employed in the past to investigate the kinetic properties of ion channel unitary events from recordings with low signal-to-noise ratios (*Anderson and Stevens, 1973*; *Zingsheim and Neher, 1974*). For *autocorrelation*, a post-acquisition Gaussian digital filter was used at one fourth of the acquisition frequency. The background signal was averaged and the standard deviation (SD) calculated. *Unitary fluctuation analysis*: when analyzing the fluctuations of fluorescence, we aim to avoid false-positive signals and therefore defined 2.5 SD ($\mu \pm 2.5\ \sigma = 0.987$) of the background signal (or the equivalent base level) as a threshold to discriminate the two regimes (*Peterson and Harris, 2010*), basal level ($L_0$) and higher signal level ($L_1$). We reasoned that fluorescence fluctuations, intrinsic to the probe, might interfere with our calculations; Soluble coumarin deposited on a glass coverslip showed 12% of positive transitions, a value we linearly subtracted from all the individual points taken from the recorded mutants (Y671$^{coum}$ or W426$^{coum}$), so that a new corrected dataset having the apparent $P_{L1}$ was obtained after subtraction. The fluorescence of the single emitter bleaches in less than 1 min of continuous illumination at 75% laser power. Normalization of optical data was performed by dividing signal amplitude by the maximal signal obtained (F/F$_{max}$). Autocorrelation, idealization, and the analysis of unitary transitions was performed with pClamp 8.0 software (RRID:SCR_011323).

## HEK-293T electrophysiology

Whole-cell currents were obtained from transiently transfected HEK-293T cells. Gigaseals were formed by using 2–4 M$\Omega$ borosilicate pipettes (Warner Instruments, Hamden, CT). Junction potentials (6–8 mV) were corrected for CsF/NaCl solutions. Seal resistance was 2–3 G$\Omega$ in all cases. Series resistance and cell capacitance were analogically compensated for directly on the amplifier. The mean maximum voltage drop was about 4 mV. Whole-cell voltage clamp was performed, and macroscopic currents from voltage steps were acquired at 20 kHz and filtered at 10 kHz. Ramp protocols were acquired at 10 kHz and filtered at 5 kHz. Currents are presented in terms of densities. Data were acquired using an Axopatch 200B and a digidata 1320 (Molecular Devices Inc., Sunnyvale, CA). The acquisition and basic analysis of the data were performed with pClamp 8.0 software.

## Statistics

*Single photon counting*. Individual cells on one day of transfection contribute with several puncta, which are considered as replicates. Therefore the histogram presented in *Figure 1g* corresponds to n = 1 but comes from the average of several individual histograms (in this case, five on each curve). The final statistical procedure was done by comparing the mode of the different populations, n = 4 for each condition. A *p*-value of 0.05 was considered significant when comparing the modes of the photon count (Mann-Whitney test). *Idealization and unitary transitions*. The idealization was performed first by checking that the base level was similar to the background noise, then setting a defined threshold on two standard deviations of background noise average. The comparison between the dwell times was performed in paired experiments (0 versus 3 uM, n = 11). The full data set for the dose-response curve was analyzed using one-way ANOVA. In general, when averaged, data are shown with the correspondent SEM. The statistical analysis was computed in Microcal OriginPro ver9 (OriginLab) (RRID:SCR_002815). Figures were prepared using Microcal OriginPro ver9. VMD (structural data), and ImageJ (RRID:SCR_003070). Source data is available through the Dryad repository (https://doi.org/10.5061/dryad.1kc2c)

## Acknowledgements

X Steinberg is a MECESUP and CONICYT fellow. This work was supported by FONDECYT grants 1110906 and 1151430 (SB) and by Anillo Científico ACT-1401 (SB). MiNICAD is a Millennium Nucleus supported by Iniciativa Científica Milenio, Ministry of Economy, Development and Tourism: PCCI12023, DRI-CONICYT/CONACYT (SB and LDI). CAA is a supported by the National Institute of Health (NIH) (GM106569, GM122420), is an American Heart Association Established Investigator (5EIA22180002), and is a member of the Membrane Protein Structural Dynamics Consortium (NIH GM087519). LDI is supported by DGAPA-PAPIIT-UNAM grant IN209515 and Fronteras de la Ciencia-CONACYT Grant 77. SB is part of CISNe-UACh and the UACh Program for Cell Biology. VC acknowledges support from the NIH (GM093290) and the National Science Foundation (ACI-1614804).

## Additional information

### Funding

| Funder | Grant reference number | Author |
|---|---|---|
| National Science Foundation | ACI-1614804 | Vincenzo Carnevale |
| National Institutes of Health | R01GM093290 | Vincenzo Carnevale |
| National Institutes of Health | S10OD020095 | Vincenzo Carnevale |
| Consejo Nacional de Ciencia y Tecnología | PCCI12023 | Leon Islas Sebastian E Brauchi |
| National Institutes of Health | R01GM106569 | Christopher A Ahern |
| National Institute of General Medical Sciences | GM087519 | Christopher A Ahern |
| American Heart Association | 5EIA22180002 | Christopher A Ahern |
| Fondo Nacional de Desarrollo Científico y Tecnológico | 1110906 | Sebastian E Brauchi |
| Comisión Nacional de Investigación Científica y Tecnológica | ACT-1401 | Sebastian E Brauchi |
| Fondo Nacional de Desarrollo Científico y Tecnológico | 1151430 | Sebastian E Brauchi |
| Fronteras de la Ciencia-CONACYT | 77 | Leon Islas |

The funders had no role in study design, data collection and interpretation, or the decision to submit the work for publication.

### Author contributions

Ximena Steinberg, Conceptualization, Formal analysis, Investigation; Marina A Kasimova, Jason D Galpin, Investigation, Methodology; Deny Cabezas-Bratesco, Ernesto Ladron-de-Guevara, Formal analysis, Investigation; Federica Villa, Resources, Software, Methodology; Vincenzo Carnevale, Conceptualization, Resources, Formal analysis, Supervision, Funding acquisition, Investigation, Methodology, Writing—original draft, Project administration, Writing—review and editing; Leon Islas, Conceptualization, Formal analysis, Supervision, Funding acquisition, Investigation, Methodology, Writing—original draft, Writing—review and editing; Christopher A Ahern, Conceptualization, Resources, Supervision, Funding acquisition, Methodology, Writing—original draft, Project administration, Writing—review and editing; Sebastian E Brauchi, Conceptualization, Formal analysis, Supervision, Funding acquisition, Investigation, Methodology, Writing—original draft, Project administration, Writing—review and editing

## Author ORCIDs

Federica Villa iD http://orcid.org/0000-0002-9840-0269
Vincenzo Carnevale iD http://orcid.org/0000-0002-1918-8280
Christopher A Ahern iD http://orcid.org/0000-0002-7975-2744
Sebastian E Brauchi iD http://orcid.org/0000-0002-8494-9912

## Decision letter and Author response

Decision letter https://doi.org/10.7554/eLife.28626.sa1
Author response https://doi.org/10.7554/eLife.28626.sa2

## Additional files

### Supplementary files

• Transparent reporting form

### Data availability

The following dataset was generated:

| Author(s) | Year | Dataset title | Dataset URL | Database and Identifier |
|---|---|---|---|---|
| Steinberg X, Kasimova M, Cabezas-Bratesco D, Galpin J, Guevara ELd, Villa F, Carnevale V, Islas L, Ahern C, Brauchi S | 2017 | Data from: Conformational dynamics in TRPV1 channels reported by an encoded coumarin amino acid | http://dx.doi.org/10.5061/dryad.1kc2c | Available at Dryad Digital Repository under a CC0 Public Domain Dedication, 10.5061/dryad.1kc2c |

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
