## [Decision Letter]

Thank you for submitting your article "Conformational dynamics in TRPV1 channels reported by an encoded coumarin amino acid" for consideration by *eLife*. Your article has been favorably evaluated by Richard Aldrich (Senior Editor) and three reviewers, one of whom, László Csanády (Reviewer #1), is a member of our Board of Reviewing Editors.

The reviewers have discussed the reviews with one another and the Reviewing Editor has drafted this decision to help you prepare a revised submission.

This is a highly ambitious study pioneering in two techniques: exploiting engineered unnatural aminoacids as a means for targeted introduction of a small fluorescent probe into a protein, and using single-molecule imaging to track in real time conformational changes of a membrane protein in a living cell. The authors apply these techniques to address potential gating-associated motions of a residue (Y671) in the vicinity of the selectivity filter of the TRPV1 cation channel. By incorporating hydroxyl-coumarin into position 671 the authors show that this residue is exposed to a different local environment in the presence of the activating ligand capsaicin and interpret this to indicate that Y671 is more water exposed (dimmer) in the closed state. They conclude that channel opening is accompanied by a conformational change of the Y671 side chain, which decreases its solvent exposure. The results suggest that the TRPV1 selectivity filter plays a role in gating, and illustrate the power of optical recordings for studying protein conformational dynamics at a single-molecule level.

The experiments are carefully designed and carried out, and seem to mostly support the conclusions. However, there are a number of conceptually important points in the manuscript that are unclear to the reviewers. These will need to be clarified before the manuscript can be published. Specific guidelines are provided below.

Essential revisions:

1) Autocorrelation time constant:

"changed the slow time constant from 1360+/-287 ms to 580+/-182 ms […] reporting on an agonist-induced increase in tau. Such a result can be interpreted in multiple ways: i) the stabilization of the open state after ligand binding (i.e. longer bursts of activity)…"

There is double confusion here. First, tau decreases, rather than increases, upon agonist addition. Second, the autocorrelation time constant is the inverse of the sum of the opening and closing rate (for a two-state channel tau=1/(kco+koc)). Therefore, a lengthening of open times (i.e., a decrease in koc) would prolong, rather than shorten this time constant. Also, please comment on how the autocorrelation time constant in the presence of capsaicin (>500 ms, Figure 3C) agrees with the current activation time constant seen in electrophysiological measurements (t_1/2_ of capsaicin activation is on the order of 100 ms according to Yao, Liu and Qin, BJ, 2010). Is the component (1364ms) at 0 cap due to spontaneous openings? If so, how does this value correlate with the spontaneous gating kinetics of TRPV1.

2) Dwell-time analysis:

Please comment on the reliability of the threshold analysis shown in Figure 3B. The choice of a threshold at 2.5 SD of the background signal seems arbitrary. It appears that if the threshold had been chosen at e.g., 2.3 SD, then many additional "openings" would have been identified (although for a Gaussian-distributed background noise the area under the tail beyond 2.3 SD would still be expected only ~1%). Is there anything to be said about an optimal choice of the threshold, and about the specificity vs. sensitivity of the threshold approach?

3) Figure 3E-F:

Please clarify what the difference is between the plot shown with black symbols in Figure 3E and the plot shown with gray symbols in Figure 3F (both fitted with orange lines), apart from normalization? The plot in 3E reports a less-than-2-fold, that in 3F an ~3-fold, stimulation by saturating capsaicin concentrations.

How can the only 2x (3x?) increase in PL1 upon exposure to saturating capsaicin, suggested by the analysis of the optical recordings (Figure 3E-F, plots fitted with orange lines), be reconciled with the ~10x increase in Po measured in electrophysiological recordings (Figure 3F, plot fitted with blue line)? Is the PL1 of ~0.2 in the absence of capsaicin related to the "intrinsic blinking" of the dye (Discussion, third paragraph)? But there it is said that this "background open probability" was subtracted from the data. Do the plots shown in Figures 3E-F represent such corrected, or uncorrected data? – This should be clearly indicated in the figure legend.

4) Electrophysiology:

Why was the capsaicin concentration dependence of TRPV1 currents (Figure 3F) obtained at +140 mV, whereas the optical measurements on intact cells clearly reflect behaviour at negative (physiological) membrane potentials? Given the voltage dependence of TRPV1 gating, the EC50 for capsaicin activation is expected to be dramatically different at +140 mV vs. Vm<0.

5) The fluorescence data traces for the control W426 control position are not shown. Please make these control traces and photon counts a prominent part of the main paper so the reader can compare the behavior of control positions with both types of imaging.

6) The molecular simulation was done with wild-type, not coumarin-carrying channels. Y671 is a critical residue (e.g., cysteine replacement severely disrupts channel function), so it is hard to predict how the local pore structure might be perturbed by the coumarin -> Tyr substitution. For that reason, it is uncertain to what extent the simulation results can be extrapolated to mutant channels. Thus, this part might be better positioned in the Discussion section.

7) It should be pointed out that the proposed hydration model is not conclusive: the data support a change in the local environment around Y671, but many other factors could be involved that may contribute to these changes by perturbing local polarity, including movement of the pore helix with associated changes in the direction of its dipole moment. A more thorough discussion of this issue would improve the paper.

8) The methodology (coumarin incorporation and optical measurement) should be described in more detail. For example, what is the efficiency of incorporation of non-natural amino acid? How about the specificity (i.e., off-target incorporation)? How bright is the fluorescence (relative to conventional dyes or FPs)? How is the 7/1 expression ratio controlled? How long does the dye last for recording before it is bleached in the single-molecule recording mode?

---

## [Author Response]

Essential revisions:1) Autocorrelation time constant:"changed the slow time constant from 1360+/-287 ms to 580+/-182 ms […] reporting on an agonist-induced increase in tau. Such a result can be interpreted in multiple ways: i) the stabilization of the open state after ligand binding (i.e. longer bursts of activity)…"There is double confusion here. First, tau decreases, rather than increases, upon agonist addition. Second, the autocorrelation time constant is the inverse of the sum of the opening and closing rate (for a two-state channel tau=1/(kco+koc)). Therefore, a lengthening of open times (i.e., a decrease in koc) would prolong, rather than shorten this time constant. Also, please comment on how the autocorrelation time constant in the presence of capsaicin (>500 ms, Figure 3C) agrees with the current activation time constant seen in electrophysiological measurements (t_1/2_ of capsaicin activation is on the order of 100 ms according to Yao, Liu and Qin, BJ, 2010). Is the component (1364ms) at 0 cap due to spontaneous openings? If so, how does this value correlate with the spontaneous gating kinetics of TRPV1.

We have to apologize and thank the reviewers for this critique. After making a closer (re)inspection of the data based on this comment we detected two relevant numerical errors on our part that were overlooked during preparation of the data:

a) When moving information from acquisition to analysis, the time base of the correlated data was copied with a 10-fold error.

b) When plotting the data, columns were labeled incorrectly as the reported time constants were inverted and didn’t make sense – as correctly pointed out by the reviewers.

In response, the data were re-analyzed from scratch and although the newly calculated values obtained are not exactly coincident with the values obtained for the dwell times (calculated from unitary transitions), they are in the same order of magnitude and retain the same relation (2-fold difference) of those observed in the originally submitted manuscript. Moreover, the data is now in close agreement to F. Qin previous report (cited in the text). The differences are commented now in the text (subsection “Activation kinetics obtained from the optical signal”, first paragraph).

2) Dwell-time analysis:Please comment on the reliability of the threshold analysis shown in Figure 3B. The choice of a threshold at 2.5 SD of the background signal seems arbitrary. It appears that if the threshold had been chosen at e.g., 2.3 SD, then many additional "openings" would have been identified (although for a Gaussian-distributed background noise the area under the tail beyond 2.3 SD would still be expected only ~1%). Is there anything to be said about an optimal choice of the threshold, and about the specificity vs. sensitivity of the threshold approach?

The threshold of 2.5SD, which we agree is arbitrary, is commonly used and in this example, follows the analysis from Peterson and Harris (DOI: 10.1021/ac901710t) based on the population size.

To clarify the analysis, first, we assumed the background noise was predicted by a Gaussian distribution, and then aimed for a statistical criterion identifying less than 2% false positives (µ ± 2.5 σ = 0.987).

Further, when the different datasets on the manuscript were analyzed together, they all point out to the same direction (i.e. a role of the upper pore on gating). Moreover, we aimed at steady state analysis for the case of fluctuations of fluorescence, and the match between electrophysiology and imaging was compelling enough not to explore further on the thresholds used for detection. We reasoned that lowering the threshold would just affect the base line of the imaging curve; therefore lower probabilities are the least sensitive part of our measurements.

3) Figure 3E-F:Please clarify what the difference is between the plot shown with black symbols in Figure 3E and the plot shown with gray symbols in Figure 3F (both fitted with orange lines), apart from normalization? The plot in 3E reports a less-than-2-fold, that in 3F an ~3-fold, stimulation by saturating capsaicin concentrations.How can the only 2x (3x?) increase in PL1 upon exposure to saturating capsaicin, suggested by the analysis of the optical recordings (Figure 3E-F, plots fitted with orange lines), be reconciled with the ~10x increase in Po measured in electrophysiological recordings (Figure 3F, plot fitted with blue line)? Is the PL1 of ~0.2 in the absence of capsaicin related to the "intrinsic blinking" of the dye (Discussion, third paragraph)? But there it is said that this "background open probability" was subtracted from the data. Do the plots shown in Figures 3E-F represent such corrected, or uncorrected data? This should be clearly indicated in the figure legend.

The optical data on Figure 3E is non-corrected and the optical data on Figure 3F is corrected and normalized by its maximum. This is now indicated in the figure legend.

The 2 to 3 fold change is a result of the subtraction of dye’s intrinsic blinking.

Regarding the relation between Po measured in electrophysiological recordings and the optically obtained PL1, we prefer to compare the midpoint of the fitted function rather than absolute open probability. This has been chosen because despite the insights gained with the current approach, we are unable to pinpoint the exact mechanism affecting the spectroscopic properties of the dye in that particular environment. Additional experiments and methods will be needed to address this issue.

4) Electrophysiology:Why was the capsaicin concentration dependence of TRPV1 currents (Figure 3F) obtained at +140 mV, whereas the optical measurements on intact cells clearly reflect behaviour at negative (physiological) membrane potentials? Given the voltage dependence of TRPV1 gating, the EC50 for capsaicin activation is expected to be dramatically different at +140 mV vs. Vm<0.

Our mistake – the recordings were taken at -70 mV, not +140. This typo is now corrected in the text. TRPV1 is an allosteric channel; we should expect an effect when the EC50 is measured at negative or positive holding potentials. However, in our hands just a modest left shift on the EC50 (0.3 to 0.7 μm, +140 versus -60 mV) is accompanied by a ~15% increase of basal Po at high positive potentials (+140 mV). This was previously reported in (Jendryke et al. 2015; DOI: 10.1038/srep22007).

5) The fluorescence data traces for the control W426 control position are not shown. Please make these control traces and photon counts a prominent part of the main paper so the reader can compare the behavior of control positions with both types of imaging.

We have now prepared Figure 1—figure supplement 2 with examples of crude traces of photon counts over time corresponding to 671 data. Together with this new figure, we have also improved Figure 3—figure supplement 1 to now contain unprocessed recordings of photon counts over time, and the corresponding fluorescence fluctuations for 671 in Figure 3b. Our goal with this exercise, consistent with the *eLife* ethos, is to enable readers to compare and evaluate all raw critical data (i.e. photon counts and fluorescence fluctuations) for 671 and 426 control.

6) The molecular simulation was done with wild-type, not coumarin-carrying channels. Y671 is a critical residue (e.g., cysteine replacement severely disrupts channel function), so it is hard to predict how the local pore structure might be perturbed by the coumarin -> Tyr substitution. For that reason, it is uncertain to what extent the simulation results can be extrapolated to mutant channels. Thus, this part might be better positioned in the Discussion section.

We thank the reviewers for this comment that has provided motivation to engage in additional molecular modelling experiments. Specifically, we concur entirely with the reviewers’ assessment that the previous version of the manuscript could have misleadingly suggested a direct comparison between simulations and experiments. Therefore, we extensively edited the manuscript to clarify that the goal of our computational analysis was to show how molecular simulations independently highlighted Y671 as a residue crucially involved in the close to open transition and undergoing significant conformational changes that affect its hydration and H-bonding properties. We believe that these observations lend some confidence to the hypothesis that the structural changes reported by coumarin are physiological and informative about the gating process. Importantly, to make the connection with experiments more direct, we now include the results from two new simulations (open and closed states) performed on a mutated channel bearing a coumarin side chain at position 671. Thus, we included an additional Figure (5) which lays out the salient features of the open and close states, for the wild type and the mutant, together with their proposed mechanistic consequences.

Because of the additional data and figure, moving the computational results to the “Discussion” section would now make the latter considerably longer and, in our opinion, more difficult to read. We hope, though, that the reviewers will agree that these clarified Results and Discussion sections provide an overall improvement of the description of the data.

7) It should be pointed out that the proposed hydration model is not conclusive: the data support a change in the local environment around Y671, but many other factors could be involved that may contribute to these changes by perturbing local polarity, including movement of the pore helix with associated changes in the direction of its dipole moment. A more thorough discussion of this issue would improve the paper.

The reviewers are correct, although the interpretations of the results are helped by the fact that capsaicin agonist activity on TRV1 channels is well documented, there is no reason to just comment on one of the possible factors that affects coumarin’s spectroscopic properties. These factors are now commented in the Discussion section.

8) The methodology (coumarin incorporation and optical measurement) should be described in more detail. For example, what is the efficiency of incorporation of non-natural amino acid? How about the specificity (i.e., off-target incorporation)? How bright is the fluorescence (relative to conventional dyes or FPs)? How is the 7/1 expression ratio controlled? How long does the dye last for recording before it is bleached in the single-molecule recording mode?

We have now included these details and illumination conditions in detail within the Materials and methods. For the efficiency of the non-natural amino acid incorporation we provide details in the Materials and methods section and a reference containing the data supporting the text included (Steinberg et al., BioRxIv, 2016). Briefly stated, the incorporation efficiency is comparatively low to other evolved synthetases. This is likely due to the fact that the synthetase used here was rationally evolved through iterative rounds of molecular docking and mutagenesis, as described in the BioRxiv paper and the Materials and methods sections. However, for its application here, the relatively low incorporation efficiency and resulting sparse labeling is an advantage for the purpose of imaging single emitters.